# Rapid and sensitive detection of selective 1,2-diaminobenzene based on facile hydrothermally prepared doped $Co_3O_4/Yb_2O_3$ nanoparticles

**Mohammed M. Rahman** *

Department of Chemistry, King Abdulaziz University, Faculty of Science, Jeddah, Saudi Arabia

* mmrahman@kau.edu.sa, mmrahmanh@gmail.com

**Data Availability Statement:** All relevant data are within the paper.

**Funding:** This work was funded by the Deanship of Scientific Research (DSR), King Abdulaziz University, Jeddah, under grant No. 130-019-

## Abstract

In this approach, the performance of a newly developed sensor probe coated with low-dimensional $Co_3O_4/Yb_2O_3$ nanoparticles (NPs) in rapidly detecting 1,2-diaminobenzene was evaluated by an electrochemical technique. The sensor probe was fabricated by depositing a very thin layer consisting of synthesized $Co_3O_4/Yb_2O_3$ NPs using a 5% Nafion conducting binder onto a glassy carbon electrode (GCE). The facile hydrothermally prepared $Co_3O_4/Yb_2O_3$ NPs were totally characterized by conventional methods such as FTIR, UV-vis, TEM, XPS, EDS, and XRD analyses. The fabricated chemical sensor probe was found to exhibit long-term activity, stability in electrochemical response, good sensitivity ($5.6962\ \mu A\mu M^{-1}cm^{-2}$), lowest detection limit ($0.02\pm0.001$ pM), and broad linear dynamic range (0.1 pM to 0.01 mM). The observed performances suggest that the newly introduced sensor could play an efficient role in detecting 1,2-diaminobenzene especially in healthcare and environmental applications on a broad scale.

## Introduction

Generally, 1,2-diamino benzene (1,2-DAB) is an organic amine and industrially very important chemical intermediate. It has extensive usage in producing medicine, auxiliaries, pesticides, dyes, pigments, photosensitive materials and, so on [1]. 1,2-DAB has also been reported [2, 3] as a carcinogenic and mutagenic substance causing hazardous effects in the human body through inhalation, ingestion, eye contact and, so on. Heavy exposure to 1,2-DAB can often cause cancer and damage liver, respiratory, and digestive systems of the human body. The ACGIH (American Conference of Governmental Industrial Hygienists) has identified 1,2-DAB as a serious hazardous substance that causes environmental pollution [4]. Therefore, it drives the need to search for a sustainable method to detect 1,2-DAB efficiently. Currently, various numerical methods including liquid chromatography [5–7], spectrophotometry [8, 9], capillary electrolysis [10], etc are widely used in detecting 1,2-DAB. The technical advantages of these methods include good sensitivity, high selectivity, and long-term stability. However, these approaches have different drawbacks including complicated detection process,

D1433. The authors, therefore, acknowledge with thanks DSR technical and financial support.

specialized instrumentation, laborious, high cost, and so on [11]. To ensure the real-time detection of this hazardous toxin, the electrochemical method is most effective. Due to the attractive electrical and optical properties, the various nanostructured metal oxides such as Pd-Rh nano-frames [12], Au@Pt core/shell nanorods [13], $Fe_3O_4$ magnetic nanoparticles [14], and tungsten carbide nanorods [15] have been recognized as potential sensing elements in the chemical sensor for detecting 1,2-DAB. The chemical sensor fabricated by $Fe_3O_4$ doped functionalized multiwall carbon nanotubes composite ($Fe_3O_4$@f-MWCTNs) is reported for its efficient performance with the sensitivity of 2.8002 $mAmM^{-1}cm^{-2}$ as well as the detection limit of 50.0 μM [16]. Similarly, Fe-MIL-88-$H_2O_2$-OPD based DAB (diaminobenzene) chemical is reported for its good performance with 50.0 nM– 30.0 μM LDR and 46.0 nM DL [17].

In the present study, $Co_3O_4$/$Yb_2O_3$ NPs/Binder/GCE based on a novel electrochemical sensor is introduced for detecting 1,2-DAB in an optimized buffer system successfully. Firstly, the slurry of $Co_3O_4$/$Yb_2O_3$ NPs in ethanol is prepared and then deposited a very thin layer onto GCE with an additive of 5% Nafion binder. This fabricated sensor is applied to detect selective 1,2 DAB through the implementation of a reliable electrochemical method by using the electrochemical approach. It is worth mentioning that the development of $Co_3O_4$/$Yb_2O_3$ NPs based GCE sensor seems to be promising technology especially for detecting toxicant to the safety of the environment.

## Experimental procedures

### Materials and experimental methods

The required inorganic salts of cobalt (II) chloride ($CoCl_2$), ytterbium chloride ($YbCl_3$) and ammonium hydroxide ($NH_4OH$) for preparing the required nano-materials supplied by SAC (Sigma Aldrich Company), USA. The other required ingredients including analytical grade 1,2-diamino benzene (1,2-DAB), 3-methyl aniline (3-MA), 3-chlorophenol (3-CP), 2,4-dinitrophenol (2,4-DNP) benzaldehyde (BH), pyridine (P), 3-methoxy hydrazine (3-MPHyd), ammonium hydroxide (AH), ethanol (E), tetrahydrofuran (3-THF), 5% Nafion, monosodium phosphate ($NaHPO_4$) and disodium phosphate ($Na_2PO_4$) were also purchased from the Sigma Aldrich Company. The prepared NPs were analyzed by XRD for identifying its crystalline phases. FESEM (JSM7600F, JEOL, Japan) was used to investigate elemental analysis, molecular arrangement, shape and size as well as morphological structure. The XPS (Thermo scientific, K-α1 1066) analysis was conducted to examine the binding energy among Co, Yb, O and their states of oxidation. The details of the XPS excitation radiation source are as follow: Al K-α1, 300.0 μm beam spot, 200.0 eV pass energy, $10^{-8}$ Torr pressure. The prepared $Co_3O_4$/$Yb_2O_3$ NPs were also examined by UV-vis spectrophotometer and FTIR (Thermo scientific NICOLET iS50, Madsion, USA). During fabricating the chemical sensor probe, an ethanolic slurry of calcined $Co_3O_4$/$Yb_2O_3$ NPs and 5% Nafion were used to develop a thin layer on GCE. It was then employed for detecting 1,2-DAB in aqueous medium. The Keithley electrometer (6517A; Purchased from the USA) was utilized to conduct electrochemical analysis through electrochemical technique.

### Hydrothermal process to prepare $Co_3O_4$/$Yb_2O_3$ NPs

The facile hydrothermal process is applied to prepare the $Co_3O_4$/$Yb_2O_3$ NPs. The hydrothermal is an efficiently used method to synthesis nano-doped material. This process basically consists of three major steps. These include (i) hydroxides co-preparation in aqueous medium, (ii) precipitation drying, and then (iii) calcining in the muffle furnace. For executing the experimentation, ytterbium chloride ($YbCl_3$) and cobalt chloride ($CoCl_2$) are dissolved in 100.0 mL de-ionized water. This mixing process is done in 250.0 mL conical flask. Since the

hydrothermal process works in an alkaline medium, 0.1 M $NH_4OH$ solution is added dropwise for adjusting the pH at 10.5 under continuous magnetic stirring. The ions $Co^{+2}$ and $Yb^{+3}$ are co-precipitated with the state of $Co(OH)_2/Yb_3(OH)_2$. The formed metal hydroxides are separated and then carefully washed with di-ionized water as well as ethanol. Then the separated precipitate is dried in an oven at a temperature of 105 ˚C. Subsequently, the dried precipitate is placed inside the furnace to calcine at 500 ˚C temperature for 6h. In the presence of oxygen, this calcination process transforms the metal hydroxides into $Co_3O_4/Yb_2O_3$ NPs. The obtained sample is then ground to particles at nano level by using a motor. The possible reaction schemes are listed below:

### Reactions in aqueous medium

$$NH_4OH_{(s)} \rightarrow NH_4^+ + OH^-_{(aq)} \tag{1}$$

$$CoCl_{2(s)} \rightarrow Co^{2+}_{(aq)} + 2Cl^-_{(aq)} \tag{2}$$

$$YbCl_{3(s)} \rightarrow Yb^{3+}_{(aq)} + 3Cl^-_{(aq)} \tag{3}$$

$$Co^{2+}_{(aq)} + Yb^{3+}_{(aq)} + 5OH^-_{(aq)} + nH_2O \rightarrow Co(OH)_2/Yb(OH)_{3(s)}.nH_2O \downarrow \tag{4}$$

### Reactions occurred in muffle furnace

$$Co(OH)_2/Yb(OH)_{3(s)} + O_2 \rightarrow Co_3O_4/Yb_2O_3 + H_2O_{(v)} \tag{5}$$

Due to the presence of several metal ions in the reaction medium, the precipitation of metal ions as metal hydroxides are dependent on the product solubility (Ks) of the corresponding metal hydroxides. It is noted that the values of product solubility (Ks) for $Yb(OH)_3$ and Co $(OH)_3$ are $1.0x10^{-22}$ and $5.92x10^{-15}$ respectively [18]. As the addition of 0.1 M $NH_4OH$ solution dropwise in the reaction medium, the concentration of $OH^-$ is increased gradually. Therefore, $Yb(OH)_3$ starts to precipitate first due to its lower value of product solubility (Ks). It forms Yb $(OH)_3$ crystal nuclei and then starts the aggregation. However, as the pH keeps increasing, the $Co(OH)_3$ starts precipitating at a certain pH value. These precipitations are then absorbed on crystallites of $Yb(OH)_3$. Similar growth patterns of nano-materials are reported in literature [19–21]. The calcined $Co_3O_4/Yb_2O_3$ NP is then ground in a mortar to make a powder sample for full of characterization. The prepared $Co_3O_4/Yb_2O_3$ NPs are applied to detect the selective 1,2-DAB through an electrochemical approach at room temperature.

### GCE fabrication by using $Co_3O_4/Yb_2O_3$ NPs

A slurry/paste of $Co_3O_4/Yb_2O_3$ NPs was prepared by mixing ethanol in order to prepare a thin layer coated GCE. The prepared GCE was then dried at ambient temperature. A drop of 5% ethanolic emulsion of Nafion (so-called conducting binder) was added during the coating process to enhance the binding strength between GCE and used NPs. It is worth mentioning that Nafion has been reported in the literature for its capability to improve conductivity, stability, and electron transfer rate of the electrode [22, 23]. For drying the produced conducting film entirely, the fabricated electrode was put inside an oven at 34.0 ˚C temperature. At the next stage, the prepared $Co_3O_4/Yb_2O_3$ NPs/binder/GCE and 1.5 mm dia Pt-wire were used as working and counter electrode respectively for making an electrochemical cell. The 1,2-DAB

solution was used as a target analyte in the developed chemical sensor. The sensitivity (Sen) of the sensor was determined from the calibration curve that represents current vs. concentrations. Similarly, DL and LDR were determined from noise and sensitivity. The used Keithley electrometer (6517A, USA) consisting of two electrode systems supplies voltage in developing electrochemical curve. Throughout this chemical experimentation, the solution with 0.1 PBS was kept constant to 10.0 mL in glass beakers.

## Results and discussions

### Analysis of structural and optical properties

The obtained FT-IR spectra of $Co_3O_4/Yb_2O_3$ NPs are presented in Fig 1A. The FT-IR spectrum of $Co_3O_4/Yb_2O_3$ NPs is displayed two distinctive peaks at 552 and 654 $cm^{-1}$, which originate from metal-oxygen stretching vibration. The bands at 552 and 654 $cm^{-1}$ are associated with $Co^{+3}$-O and $Co^{+2}$-O vibration respectively [24, 25]. The adsorption peaks at wavenumber 1407, 1625, and 3306 $cm^{-1}$ are more likely attributed to the stretching vibration mode of O-H due to the adsorption of water from the environment [26–28]. The photo-electronic sensitivity through UV-vis analysis of prepared $Co_3O_4/Yb_2O_3$ NPs was conducted at 290–800 nm wavelength. Due to the adsorption of visible light radiant energy, the electrons of synthesized NPs are transmitted from the low-level to high-level, which resulted in UV-vis spectra [29]. The

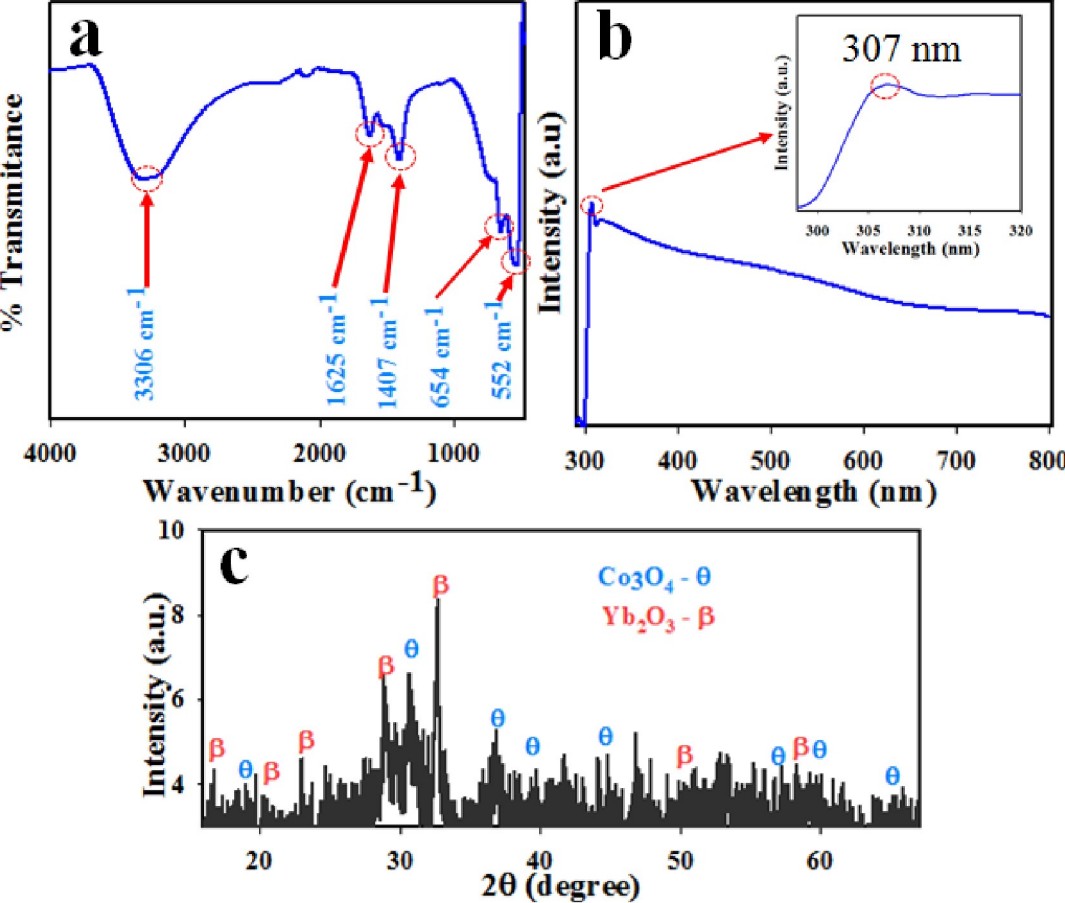

**Fig 1.** (a) FT-IR, (b) UV-vis spectrum, and (c) XRD patterns of $Co_3O_4/Yb_2O_3$ NPs for evaluation of optical and morphological properties.

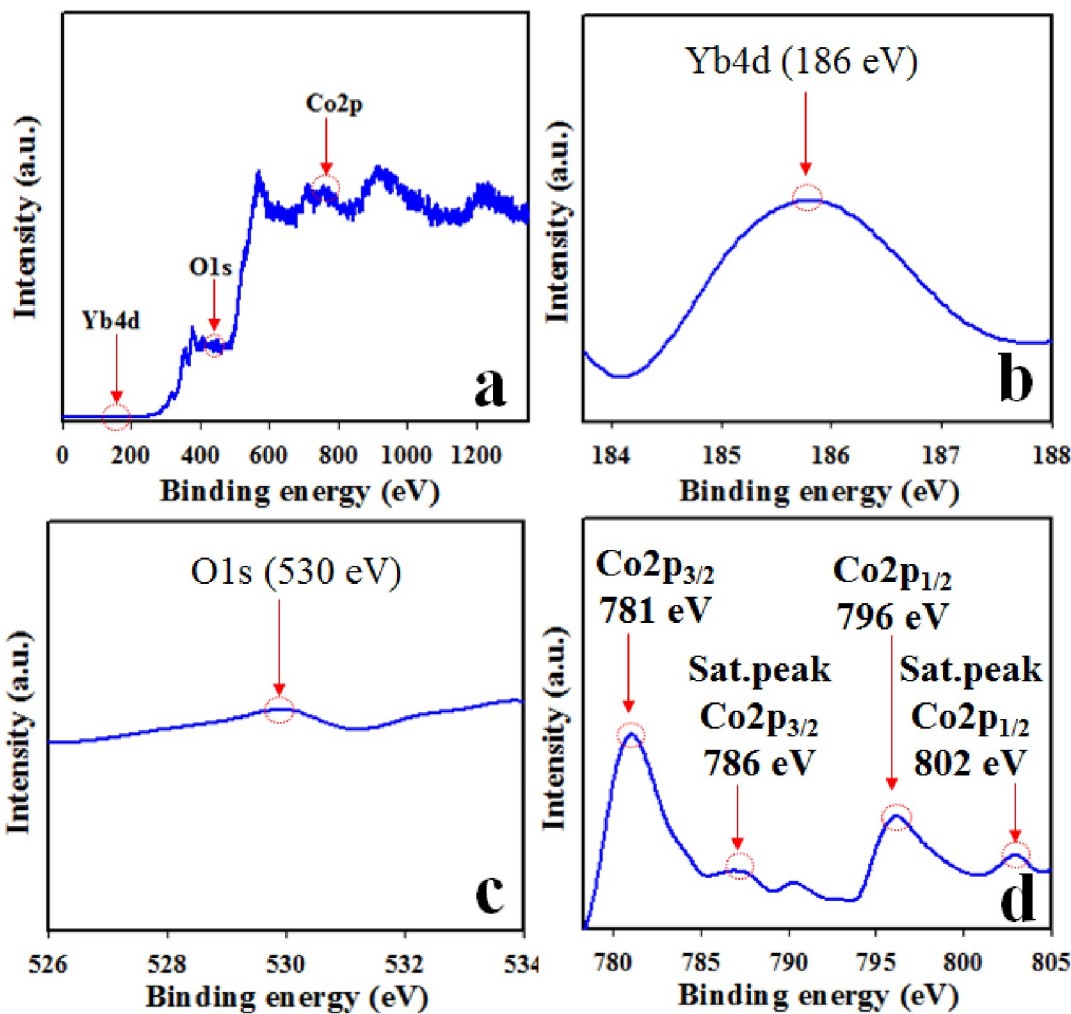

**Fig 2.** XPS analysis of $Co_3O_4/Yb_2O_3$ NPs: (a) Full range spectrum; Orbital spins of (b) Yb4d-level, (c) O1s-level, and (d) Co2p-level.

typical UV-vis spectra are presented in Fig 1B. An intense peak (as shown in the inset, Fig 1B) is observed at 307.0 nm. This could be the characteristic bond of synthesized $Co_3O_4/Al_2O_3$ NPs. The obtained UV absorption band is in good agreement with the literature [30–32]. As per Eq (6), the measured band-gap energy is found to be 4.04 eV.

$$E_g = 1240/\lambda_{max} \tag{6}$$

Where, $E_g$: band-gap energy; $\lambda_{max}$: maximum absorbed wavelength [33, 34].

The X-ray diffraction (XRD) analysis was conducted on $Co_3O_4/Yb_2O_3$ NPs in the range of 10 to 80 degrees at the scanning speed of 2 / min. The obtained XRD spectrum is presented in Fig 1C which shows the well-crystalline phases of $Co_3O_4$ and $Yb_2O_3$. The reflected crystalline peaks of $Co_3O_4$ indices as θ are (111), (220), (311), (400), (420), (440), and (511) as identified in the literature [35–37] accordingly JCPDS 42–1467. Moreover, the identified peaks of $Yb_2O_3$ indices as ß are (200), (211), (220), (222), (400), (440) and (622) as reported by the studies [38, 39] accordingly JCPDS No. 87–2374.

As per general practice, the crystal size of synthesized NPs is measured from XRD diffraction patterns by using Eq (7).

$$D = 0.91 \, \lambda/(\beta \, \cos\theta) \tag{7}$$

Here, $\lambda$: wavelength, $\beta$: width at half corresponding to the highest intense peak, and $\theta$: diffraction angle [40]. The measured crystal size by using Eq (7) is found to be 27.03 nm.

### Analysis of binding energy

The prepared NPs of $Co_3O_4/Yb_2O_3$ were also analyzed by the XPS technique. When $Co_3O_4/Yb_2O_3$ NPs are scanned using XPS, the valence electron from outer orbit transmits from low-energy level to high-energy level owing to the kinetic energy of the X-ray beam absorbed. This practice is utilized to determine the atomic composition as well as oxidation compounds present in the tested sample [41–43]. The obtained XPS spectra are presented in Fig 2A for Co2p, Yb4d, and O1s. The Co2p core level spectrum presented in Fig 2D shows two dominating peaks of Co2p3/2 and Co2p1/2 at 781 eV and 796 eV respectively along with satellite peaks. These two dominating sharp peaks can be recognized to verify the existence of $Co^{+3}$. The observed satellite peaks of Co2p3/2 and Co2p1/2 are at 786 eV and 802 eV respectively. These two peaks are more likely ascribed to the presence of $Co^{+2}$ in the prepared $Co_3O_4/Yb_2O_3$ NPs [44–48]. Thus, the presence of Co2p spin orbitals demonstrates the co-existence of Co(II) and Co(III) on the synthesized $Co_3O_4/Yb_2O_3$ NPs. The O1s peak at 530 eV as shown in Fig 3C is

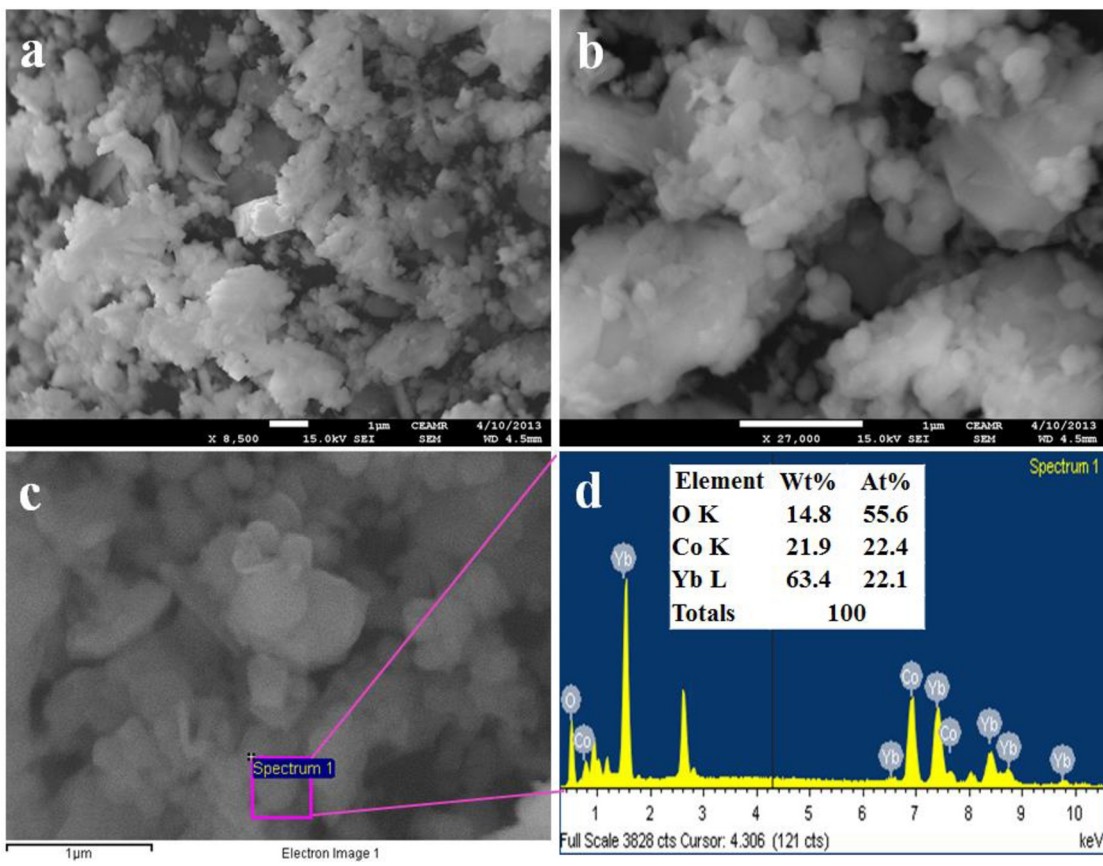

**Fig 3. Morphology and elemental analyses.** (a-b) FESEM micrographs, (c-d) EDS analysis of $Co_3O_4/Yb_2O_3$NPs.

attributed to the $O^{2-}$ in $Co_3O_4$ [49–51]. Similarly, the Yb4d peak at 186 as is ascribed to $Yb^{+3}$ – $O^{-2}$ bonds in $Yb_2O_3$ as illustrated in Fig 3B [52–54].

## Elemental and morphological analysis

Fig 3 represents the morphological and elemental analysis of prepared NPs of $Co_3O_4/Yb_2O_3$ and this analysis was carried out by the implementation of FESEM and EDS on the synthesized sample. Fig 3A and 3B clearly shows that the prepared $Co_3O_4/Yb_2O_3$ nanomaterials are spherical in shape [55–57]. The elemental analysis of $Co_3O_4/Yb_2O_3$ NPs as per EDS test presented in Fig 2C and 2D shows O 14.8%, Yb 63.4%, and Co 21.9%. Besides this, there is no other visible peak and therefore it can be concluded that the prepared $Co_3O_4/Yb_2O_3$ NPs consist of cobalt, ytterbium, and oxygen only [58, 59].

## Applications: Detection of 1,2-DAB by $Co_3O_4/Yb_2O_3$ NPs

The conductive binder (5% Nafion) was used during the fabrication of the sensor probe through the deposition of a thin layer of $Co_3O_4/Yb_2O_3$ NPs on GCE. The use of conductive binder successfully improved its conductivity, stability, and electron transfer rate. The prepared $Co_3O_4/Yb_2O_3$NPs/binder/GCE based chemical sensor was then employed to detect 1,2-DAB selectively in optimized aqueous buffer solution. In electrochemical detection of 1,2-DAB by electrochemical method, current versus potential was determined on the thin film of $Co_3O_4/Yb_2O_3$ NPs by fixing the holding time for 1 s. Scheme 1 presents the possible reaction mechanism of 1,2 DAB. It is worth mentioning that enrichment of electrons is observed during the determination of sensing performance. Thus, the conductivity of the sensing medium of 1,2-DAB is increased. As a result, amplified electrochemical responses as illustrated in Fig 5A are found to be significant with the increment of 1,2-DAB concentration. It seems that the reactive 1,2-DAB is absorbed on the fabricated working electrode (GCE coated of $Co_3O_4/$

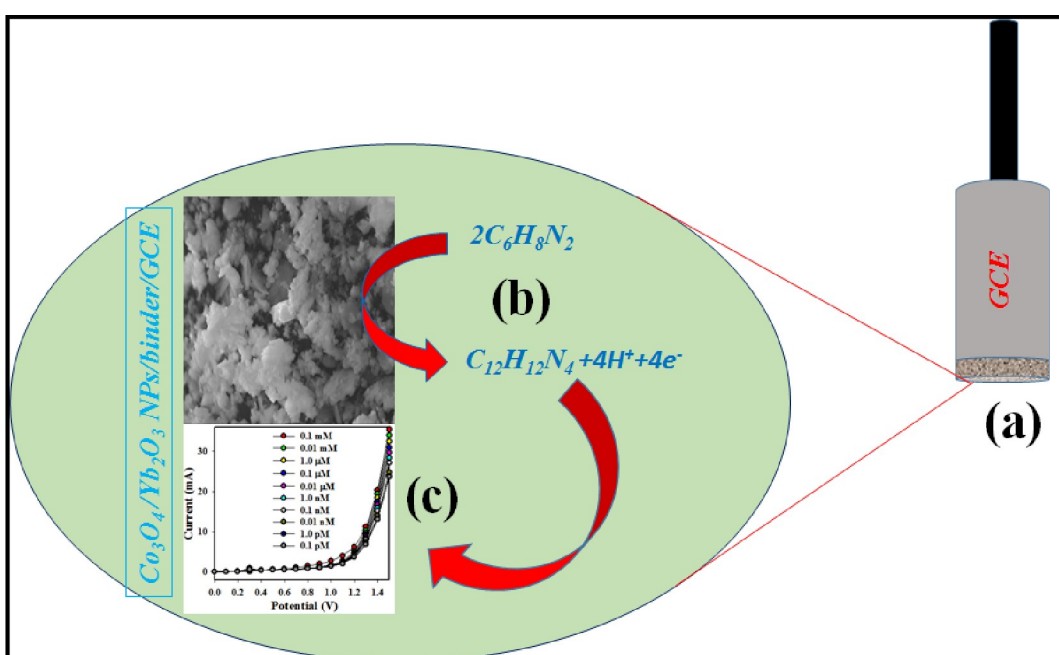

**Scheme 1. Schematic representation of the possible detection mechanism of target 1,2-DAB onto $Co_3O_4/Yb_2O_3$ NPs/ Nafion/GCE.**

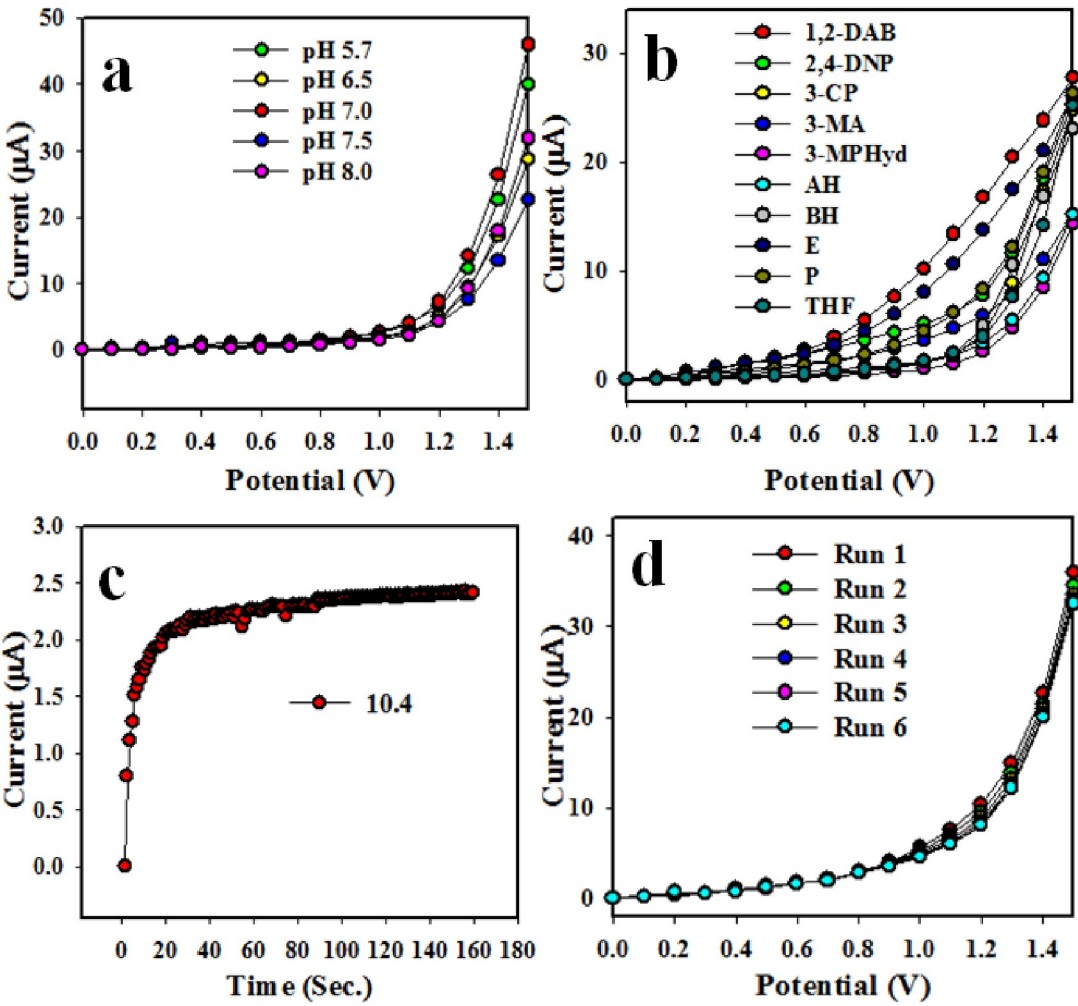

**Fig 4.** Optimization of $Co_3O_4/Yb_2O_3$ NPs/binder/GCE based 1,2-DAB chemical sensor: (a) optimization of pH, (b) level of selectivity, (c) response time and (d) degree of repeatability.

$Yb_2O_3$ NPs) surface as per reaction (8) and consequently the oxidation reactions are started. The produced electrons and hydrogen ions from oxidation of 1,2-DAB as shown in reaction (8) effectively increase the conductivity of the sensing medium [16, 60, 61]. This is why the electrochemical responses are observed to be more significant with the increase of 1,2-DAB concentrations. This demonstrates that the electrochemical responses are directly proportional to the concentration of 1,2-DAB. This is in good agreement with the results reported elsewhere [62]. The pictorial representation of the $Co_3O_4/Yb_2O_3$ NPs modified electrode in detecting of 1,2-DAB is demonstrated in Scheme 1.

The possible oxidation reaction of 1,2-DAB is presented by reaction (8) below.

$$2C_6H_8N_2 \rightarrow C_{12}H_{12}N_4 + 4H^+ + 4e^- \qquad (8)$$

The hydrothermally prepared $Co_3O_4/Yb_2O_3$ NPs are not similarly active to all the phosphate buffer systems in applied electrochemical method. Therefore, it is very important to optimize the buffer solution for obtaining the maximum electrochemical responses. The fabricated $Co_3O_4/Yb_2O_3$ NPs/binder/GCE based sensor probe was examined in different buffer solutions with pH ranging from 5.7 to 8.0 The tested chemical sensor probe was maximum

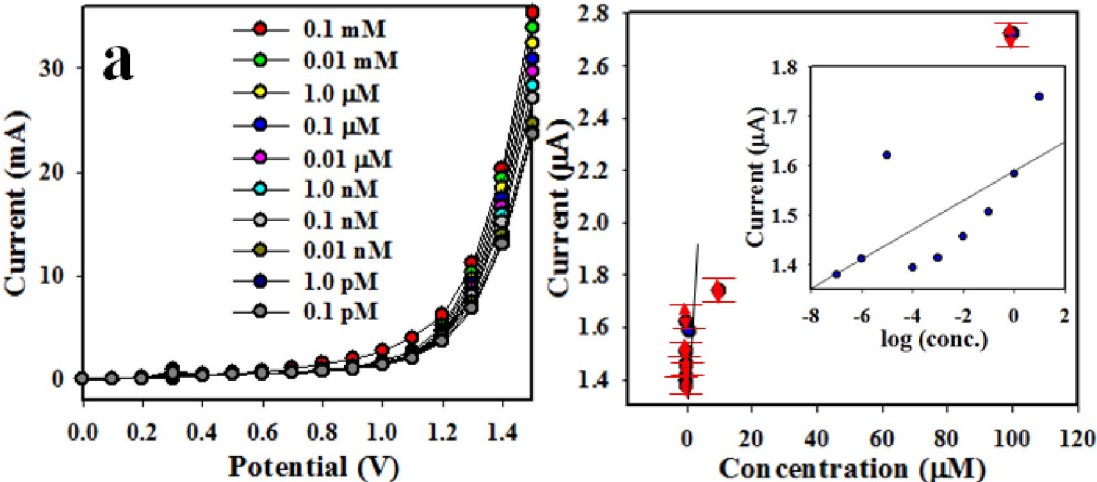

**Fig 5.** (a) Effect of concentration variation on the performance of $Co_3O_4/Yb_2O_3$ NPs/binder/GCE based 1,2-DAB chemical sensor through (a) I-V graphical presentation, (b) current versus concentration curve (Inset: Log [Conc. Of 1,2-DAB] versus current).

current at pH of 7 as it is shown in Fig 4A. This performance (pH optimization) test was conducted by using Keithley electrometer at applied potential range 0–1.5 V. The electrochemical responses were recorded for different concentrations ranging from 1.0 nM of 1,2-DAB and pH of 7. Such responses of 1,2-diamino benzene (1,2-DAB), 2,4-dinitro-phenol (2,4-DNP) 3-methyl aniline (3-MA), benzaldehyde (BH), 3-methoxy phenyl hydrazine (3-MPHyd), 3-chlorophenol (3-CP), pyridine (P), ammonium hydroxide (AH), ethanol (E) and tetrahydro-furan (THF) are illustrated in Fig 4B. Among others, 1,2-DAB was found to exhibit the maximum electrochemical responses. The key analytical performance of a sensor is denoted by the response time. This test was conducted in 1.0 nM concentration for 1,2-DAB to optimize in buffer solution. The highly applicable value of the response time as per observation from Fig 4C is 11.0 sec. Reproducibility is another important sensor performance parameter. The data on reproducibility is presented in Fig 4D after conducting the test in 1.0 nM concentration of 1,2-DAB and pH of 7. The responses were observed to be outstanding even after washing the working electrode in each trail. At the next stage, RSD (relative standard deviation) was calculated. It was found to be 2.26 at +.8 V potential.

Fig 5 shows the graphical presentation of current versus the potential for 1,2-DAB detection at the concentration ranging from 0.1 mM to 0.1 mM. The observed responses corresponding to the concentration are found to be distinguishable. The resulted plot shows the regression coefficient value ($r^2 = 0.9745$) at +1.0 V potential. The estimated sensitivity and LDR (Linear dynamic range) are found to be 5.6962 $\mu A \mu M^{-1} cm^{-2}$ and 0.1 pM– 0.01 mM respectively. The detection limit (DL) is calculated as $0.02 \pm 0.001$ pM (signal to noise ratio of 3).

Fig 5A demonstrates that the electrochemical response varies with the concentration of 1,2-DAB which is in good agreement with the finding reported in the literatures [63–65]. A few 1,2-DAB chemicals are absorbed in the very beginning of the sensing performance of 1,2-DAB with the $Co_3O_4/Yb_2O_3$ NPs/binder/GCE sensor probe. The absorption of 1,2-DAB mainly occurs when oxidation starts progressively onto the working electrode surfaces [66]. In the meantime, the analyte concentration on the surface increases and covers a larger part of the surface. It indicates a gradual increase of reaction rate on the $Co_3O_4/Yb_2O_3$ NPs/binder/GCE surface. However, the surface reaction rate and corresponding current density reach a

**Table 1. The comparative performances of different chemical sensors in detecting 1,2-DAB based on various electrode modification.**

| Modified electrode system | Detection limit (DR) | Linear dynamic range (LDR) | Sensitivity | Ref. |
|---|---|---|---|---|
| $Fe_3O_4$@f-MWCNTs | 50.00 μM | 0.6–80 μM | 2.80 $\mu A\mu M^{-1}cm^{-2}$ | [16] |
| $Co_3O_4/Yb_2O_3$ NPs /GCE | 0.02 pM | 0.1 pM ~0.01 mM | 5.69 $\mu A\mu M^{-1}cm^{-2}$ | This work |

*nM–Nanomole, μM–Micromole, pM–picomole.

steady-state level when surface coverage level approaches its equilibrium state due to the additional of the 1,2-DAB on $Co_3O_4/Yb_2O_3$ NPs/binder/GCE electrode. The experimental results presented in Fig 5B show a linear plot with the homogeneous distribution. This suggests that the proposed 1,2-DAB sensor could be applied successfully to toxin levels in buffer medium. The response time is required as 10.4 sec by the $Co_3O_4/Yb_2O_3$NPs/binder/GCE sensor probe based chemical sensing as shown in previously presented Fig 4C to reach steady-state response. Thus, it is possible to record and preserve the data within 10.4 sec. The fabricated $Co_3O_4/Yb_2O_3$NPs/binder/GCE sensor probe is found to be efficient in terms of DL and DLR selectively. Table 1 shows the comparison of chemical sensors fabricated with various nanostructured metals by electrochemical methods.

## Real sample analysis

The newly developed $Co_3O_4/Yb_2O_3$ NPs/binder/GCE based chemical sensor was justified for its applicability within the practical field through employing it in detecting 1,2-DAB with verities of real samples collected from the various sources. To confirm its wide range of applicability, the tested real samples were collected from different sources including extracts from PC baby-bottles, PC water bottles, PVC-made food packing bags, and so on. The collected samples are initially filtered. Finally, the samples are used to detect by fabricated $Co_3O_4/Yb_2O_3$ NPs/binder/GCE sensor probe using an electrochemical method in room conditions. The obtained results as presented in Table 2, are found to be quite satisfactory and acceptable.

## Conclusions

Finally, nanoparticles of $Co_3O_4/Yb_2O_3$ were prepared hydrothermally and analyzed by various conventional methods, such as UV-vis, FTIR, FESEM, EDS, XRD and XPS. A selective chemical sensor probe was successfully prepared through depositing a very thin layer of $Co_3O_4/Yb_2O_3$ NPs onto GCE. During deposition process, a drop of Nafion was added for improving its fabrication process to stick the powder NPs onto GCE surface. The fabricated sensor probe was then employed in detecting the target toxic 1,2-DAB in phosphate buffer system. The 1,2-DAB chemical sensor is exhibited satisfactory results for various analytical performing factors including detection limit, selectivity, linearity, response time, reproducibility,

**Table 2. Measured concentrations of 1,2-DAB analytes for different real samples by an electrochemical method using $Co_3O_4/Yb_2O_3$ NPs/Nafion/GCE.**

| Real sample | Observed current (μA) | | | | Average | Measured Conc. (μM) | %SD |
|---|---|---|---|---|---|---|---|
| | Reading 1 | Reading 2 | Reading 3 | Reading 4 | | | |
| Industrial effluents | 1.59 | 1.61 | 1.62 | 1.64 | 1.615 | 0.0113 | 1.29 |
| PC baby bottle | 2.84 | 2.88 | 2.95 | 2.97 | 2.910 | 0.0206 | 2.08 |
| PC water bottle | 2.26 | 2.19 | 2.14 | 2.23 | 2.205 | 0.0156 | 2.36 |
| Packing bag/PVC | 2.37 | 2.46 | 2.50 | 2.61 | 2.485 | 0.0176 | 4.00 |

*SD–Standard deviation, μM–Micromole.

repeatability, stability, and so on. The obtained sensitivity and LDR (linear dynamic range) are observed to be 5.6962 $\mu A \mu M^{-1} cm^{-2}$ and 0.1 pM—0.01 mM respectively. The DL (detection limit) is found to be 0.02 ± 0.001 pM at the point where the signal to noise ratio is 3. The test results are obtained from various real sample sources demonstrated that the fabricated sensor is highly reliable and suitable in detecting toxins by an electrochemical process on a broad scale. Thus, a sensor probe with binary doped nanostructure material is applied and useful for the safety of environmental and health care fields on a large scale.

## Author Contributions

**Conceptualization:** Mohammed M. Rahman.

**Data curation:** Mohammed M. Rahman.

**Investigation:** Mohammed M. Rahman.

**Methodology:** Mohammed M. Rahman.

**Writing – review & editing:** Mohammed M. Rahman.

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
