## [Decision Letter · Decision Letter 0]

7 Jan 2021

PONE-D-20-36128

Rapid and sensitive detection of selective 1,2-diaminobenzene based on facile hydrothermally prepared Co3O4/Yb2O3 nanoparticles for environmental remediation

PLOS ONE

Dear Dr. Rahman,

Thank you for submitting your manuscript to PLOS ONE. After careful consideration, we feel that it has merit but does not fully meet PLOS ONE’s publication criteria as it currently stands. Therefore, we invite you to submit a revised version of the manuscript that addresses the points raised during the review process.

Please submit your revised manuscript by  January 31, 2021. If you will need more time than this to complete your revisions, please reply to this message or contact the journal office at plosone@plos.org. Please include the following items when submitting your revised manuscript:

We look forward to receiving your revised manuscript.

Kind regards,

Shabi Abbas Zaidi, Ph.D.

Academic Editor

PLOS ONE

Journal Requirements:

2)  We suggest you thoroughly copyedit your manuscript for language usage, spelling, and grammar. If you do not know anyone who can help you do this, you may wish to consider employing a professional scientific editing service.  

Reviewers' comments:

Reviewer's Responses to Questions

**Comments to the Author**

1. Is the manuscript technically sound, and do the data support the conclusions?

Reviewer #1: Yes

2. Has the statistical analysis been performed appropriately and rigorously? 

Reviewer #1: Yes

3. Have the authors made all data underlying the findings in their manuscript fully available?

Reviewer #1: Yes

4. Is the manuscript presented in an intelligible fashion and written in standard English?

Reviewer #1: Yes

5. Review Comments to the Author

Reviewer #1: The idea and content of the manuscript is well. The authors have designed the selective chemical sensor embedded Nanoparticles of Co3O4/Yb2O3 for detecting the toxic 1,2-DAB in buffer system. This work is interesting and the obtained results in terms of sensitivity and linear dynamic range are satisfactory. The manuscript may be accepted for the publication, but before to acceptance the authors should incorporate the reason why considered the result on the signal to noise ratio is 3. The grammar of the manuscript has a required lot of improvement.

6. PLOS authors have the option to publish the peer review history of their article (what does this mean?). If published, this will include your full peer review and any attached files.

Reviewer #1: No

---

## [Author Response · Author response to Decision Letter 0]

18 Jan 2021

Reviewers' comments:

Reviewer's Responses to Questions

Comments to the Author

1. Is the manuscript technically sound, and do the data support the conclusions?

Reviewer #1: Yes________________________________________

2. Has the statistical analysis been performed appropriately and rigorously?

Reviewer #1: Yes________________________________________

3. Have the authors made all data underlying the findings in their manuscript fully available?

 Reviewer #1: Yes________________________________________

4. Is the manuscript presented in an intelligible fashion and written in standard English?

Reviewer #1: Yes________________________________________

5. Review Comments to the Author

Reviewer #1: The idea and content of the manuscript is well. The authors have designed the selective chemical sensor embedded Nanoparticles of Co3O4/Yb2O3 for detecting the toxic 1,2-DAB in buffer system. This work is interesting and the obtained results in terms of sensitivity and linear dynamic range are satisfactory. The manuscript may be accepted for the publication, but before to acceptance the authors should incorporate the reason why considered the result on the signal to noise ratio is 3. The grammar of the manuscript has a required lot of improvement.

Response: Thank you for the comment. From the calibration plot, it is used to calculate the sensor analytical parameters (sensitivity, linearity, and detection limit etc.) from the linear dynamic range by considering the active surface area (Surface area: 0.0316 cm2) of fabricated sensor probe. Here, for the calculation of detection limit, it is used ~3N/S, which means the signal (S) to noise (N) ratio of 3. The detection limit (DL) is calculated as 0.02 pM.

6. PLOS authors have the option to publish the peer review history of their article (what does this mean?). If published, this will include your full peer review and any attached files.

Do you want your identity to be public for this peer review? For information about this choice, including consent withdrawal, please see our Privacy Policy.

Reviewer #1: No

---

## [Editor Report · Decision Letter 1]

21 Jan 2021

PONE-D-20-36128R1

Rapid and sensitive detection of selective 1,2-diaminobenzene based on facile hydrothermally prepared doped Co3O4/Yb2O3 nanoparticles

PLOS ONE

Dear Dr. Rahman,

Thank you for submitting your manuscript to PLOS ONE. After careful consideration, we feel that it has merit but does not fully meet PLOS ONE’s publication criteria as it currently stands. Therefore, we invite you to submit a revised version of the manuscript that addresses the points raised during the review process.

Your work is suitable for PLOS One. However, there is a concern with the language of the article. We suggest you thoroughly copyedit your manuscript for language usage, spelling, and grammar. If you do not know anyone who can help you do this, you may wish to consider employing a professional scientific editing service.

Whilst you may use any professional scientific editing service of your choice, PLOS has partnered with both American Journal Experts (AJE) to provide discounted services to PLOS authors. Both organizations have experience helping authors meet PLOS guidelines and can provide language editing, translation, manuscript formatting, and figure formatting to ensure your manuscript meets our submission guidelines. To take advantage of our partnership with AJE, visit the AJE website (http://learn.aje.com/plos/) for a 15% discount off AJE services. If the PLOS editorial team finds any language issues in text that AJE has edited, the service

provider will re-edit the text for free.

- A clean copy of the edited manuscript (uploaded as the new *manuscript* file)"

ACADEMIC EDITOR: Please insert comments here and delete this placeholder text when finished. Be sure to:

**Please submit your revised manuscript by February 15, 2021**. If you will need more time than this to complete your revisions, please reply to this message or contact the journal office at plosone@plos.org. Please include the following items when submitting your revised manuscript:

We look forward to receiving your revised manuscript.

Kind regards,

Shabi Abbas Zaidi, Ph.D.

Academic Editor

PLOS ONE

---

## [Author Response · Author response to Decision Letter 1]

24 Jan 2021

Response: Dr. William Ghann (wghann@coppin.edu) at Coppin State University, USA has edited my paper.

Response: Track change file has been uploaded here.

- A clean copy of the edited manuscript (uploaded as the new *manuscript* file)"

Response: A clean copy of revised paper has been uploaded here.

---

## [Editor Report · Decision Letter 2]

26 Jan 2021

Rapid and sensitive detection of selective 1,2-diaminobenzene based on facile hydrothermally prepared doped Co3O4/Yb2O3 nanoparticles

PONE-D-20-36128R2

Dear Dr. Rahman,

We’re pleased to inform you that your manuscript has been judged scientifically suitable for publication and will be formally accepted for publication once it meets all outstanding technical requirements.

Kind regards,

Shabi Abbas Zaidi, Ph.D.

Academic Editor

PLOS ONE
---

## [Editor Report · Acceptance letter]

1 Feb 2021

PONE-D-20-36128R2 

Rapid and sensitive detection of selective 1,2-diaminobenzene based on facile hydrothermally prepared doped Co_3_O_4_/Yb_2_O_3_ nanoparticles 

Dear Dr. Rahman:

I'm pleased to inform you that your manuscript has been deemed suitable for publication in PLOS ONE. Congratulations! Your manuscript is now with our production department. 

Kind regards, 

on behalf of

Dr. Shabi Abbas Zaidi 

Academic Editor

PLOS ONE